# Comparison of Clinical Manifestations and Pathology between Kimura Disease and IgG4-Related Disease: A Report of Two Cases and Literature Review

**DOI:** 10.3390/jcm11236887

**Published:** 2022-11-22

**Authors:** Sing-Ya Chang, Chih-Chun Lee, Ming-Ling Chang, Wen-Chieh Teng, Chao-Yang Hsiao, Han-Hua Yu, Meng-Ju Hsieh, Tien-Ming Chan

**Affiliations:** 1School of Medicine, College of Medicine, Chang Gung University, Taoyuan 333, Taiwan; 2Department of Medical Education, Chang Gung Memorial Hospital, Keelung Branch, Keelung 204, Taiwan; 3Division of Gastroenterology and Hepatology, Chang Gung Memorial Hospital, Linkou Branch, Chang Gung University College of Medicine, Taoyuan 333, Taiwan; 4Division of Rheumatology, Allergy and Immunology, Chang Gung Memorial Hospital, Linkou Branch, College of Medicine, Chang Gung University, Taoyuan 333, Taiwan

**Keywords:** Kimura disease, IgG4-related disease, overlapping features, case report, Taiwan

## Abstract

Kimura disease (KD) is a rare, chronic proliferative condition presenting as a subcutaneous mass predominantly located in the head and neck region; it is characterized by eosinophilia and elevated serum IgE levels. IgG4-related disease (IgG4RD) is a fibroinflammatory condition characterized by swelling in single or multiple organs and the infiltration of IgG4 plasma cells. Herein, we presented two cases. Case 1 is a 38-year-old man with a painless mass in his right postauricular region, and Case 2 is a 36-year-old man with painless lymphadenopathy in his bilateral postauricular region. After surgical excision, they showed good recovery with no relapse. Although Cases 1 and 2 shared several overlapping pathological manifestations, there were a few differences that allowed the differentiation of KD and IgG4RD.

## 1. Introduction

Kimura disease (KD) is a chronic proliferative condition characterized by a soft subcutaneous mass in the head and neck region, focal lymphadenopathy, and lacrimal or sialic gland swelling. KD was first reported by Kim and Szeto in 1937 and was defined by Kimura et al. in 1948 [1,2]. KD is more common among young Asian men; however, individuals of another sex, ethnicity, and age may also be affected [3,4]. The histological features of KD include follicular hyperplasia; infiltration of eosinophils, lymphocytes, and plasma cells; and vascular proliferation. Serology testing usually indicates elevated serum IgE levels and peripheral eosinophilia [5]. However, its pathogenesis remains unclear. As elevated levels of IL-4, IL-5, and IL-13 were noted in a previous study, current research consensus indicates that KD is associated with autoimmune diseases or chronic inflammation [6]. The treatment options for KD include surgery or oral steroids, which can shrink the lump within a few months; however, 40% of cases may show relapse [3,4,7]. Comorbidity includes nephropathy, which occurred in 16% of patients with mesangial proliferative glomerulonephritis and minimal change disease [8].

IgG4RD is a multiorgan fibroinflammatory condition, which may affect the pancreas, major salivary glands, lacrimal glands, kidneys, bile ducts, lungs, and aorta [9]. Patients with IgG4RD are classified into four groups, and 46% of these patients belong to group 3 of head and neck limited disease and group 4 of Mikulicz disease; the characteristics of these two groups were more similar to the presentation of KD [10]. Pathologically, this disease is characterized by dense lymphoplasmacytic infiltration, storiform fibrosis, and obliterative phlebitis. However, patients with IgG4RD may also present with nonobliterative phlebitis and eosinophilic infiltration.

Since the first description of two patients with KD with IgG4(+) plasma cell infiltration by Hattori et al, there have been several reports of elevated IgG4 or IgG4/IgG levels in patients with KD [11]. However, a previous study reported elevated serum IgE levels and eosinophilia in the plasma samples of patients with IgG4RD [12]. As IL4 regulates the production of IgG4 and IgE and secretion of IL5, IL10, and IL13 by Th2 cells, some researchers have suggested that elevated serum IgE and IgG4 levels as well as eosinophilia in patients with KD and IgG4RD are associated with Th2-mediated autoimmune response [13].

This report describes two cases in which one patient was diagnosed with KD and the other with IgG4RD. The patients shared overlapping clinical and pathological manifestations, which led to difficulty in differentiating the two diseases. This report highlights the clinical, histological, and serological similarities and differences between KD and IgG4RD.

## 2. Case Report

### 2.1. Case 1

A 38-year-old Chinese man suffered from a right postauricular painless mass for several years. He did not report any pain, redness, or heat sensation. He was a smoker who quit smoking several years ago. He denied any history of systemic or allergic diseases except for allergic urticaria. Physical examination revealed a soft lump (2 cm × 2 cm) in the right postauricular region. After 1 month, the patient underwent a C-shaped incision to excise the postauricular mass under local anesthesia. Pathological analysis based on hematoxylin–eosin staining revealed lymphoid tissue with eosinophilic abscess formation, small blood vessel infiltration with fibrosis, florid lymphoid follicle hyperplasia, and eosinophilia (Figure 1), which were consistent with the manifestations of KD. Furthermore, immunohistological staining for IgG and IgG4 was performed. Within three high-power fields (HPF), we observed >50 IgG4(+) plasma cells per HPF. The ratio of IgG4(+) to IgG(+) plasma cells was >40%. Postsurgical laboratory tests revealed mild eosinophilia (5.8%, 423/μL) and elevated serum IgE (713 IU/mL) and IgG4 (193 mg/dL) levels. There was no sign of relapse during the 1-year follow-up.

### 2.2. Case 2

A 36-year-old man with bilateral postauricular painless masses for several years presented to our outpatient clinic for evaluation. He was a smoker who denied any history of systemic or allergic disease except for hepatitis C. The dimensions of the lesions were 4 cm × 2 cm on the left side and 2 cm × 2 cm on the right side. Neck ultrasound revealed multiple hypoechoic nodules, with the largest nodule of 1.7 cm in the left level II region and 1.3 cm in the right level III region. Fine needle aspiration cytology was performed simultaneously, which showed the presence of abundant lymphocytes, histiocytes, and mild neutrophil and plasma cell infiltration. High-resolution computed tomography of the temporal bone revealed subcutaneous nodular lesions in bilateral postauricular regions (dimensions, right: 15 mm × 7 mm; left: 17 mm × 12 mm) with relatively ill-defined margins and increased stranding of adjacent subcutaneous fat (Figure 2). There were no engorged vessels around these lesions.

The patient was lost to follow-up after his first visit; however, he returned to our clinic 4 years later and presented with the same but enlarged lesions (left: 8 cm × 5 cm; right: 5 cm × 4 cm) and eosinophilia (24.0%). Therefore, he underwent surgical excision. Hematoxylin–eosin staining showed that the pathological manifestations of the left lesion were interfollicular lymphoplasmacytic and eosinophilic infiltration, small blood vessel infiltration, and fibrosis in the salivary gland and lymph node tissue (Figure 3). The manifestations of the right lesion were similar to those of the left one; however, these were only observed in lymph node tissue. No malignant change or granuloma was noted. Immunohistological staining for IgG and IgG4 revealed that both lesions showed markedly increased IgG4(+) plasma cell infiltration and elevated ratios of IgG4(+) to IgG(+) plasma cells (left: >100 IgG4(+) cells per HPF, IgG4(+)/IgG(+) plasma cell ratio of >40%; right: >50 IgG4(+) cells per HPF, IgG4(+)/IgG(+) plasma cell ratio of >40%). Laboratory examination after the surgery showed elevated serum IgG4 (215 mg/dL) levels and persistent peripheral eosinophilia (19.3%, 1216/μL). The clinical, laboratory, and pathological manifestations of the patient conformed to the 2019 ACR/EULAR IgG4RD criteria [14]. He has not shown any signs of recurrence to date.

## 3. Discussion

In this report, the two patients shared several similarities, although they were diagnosed individually with KD and IgG4RD. Both patients are 30–40-year-old men with eosinophilia who showed good recovery after resection, with no relapse during the 1-year follow-up; however, they continued to present with elevated serum IgG4 levels. Their lumps were similar in size, tactility, and location, with identical pathological manifestations of eosinophilic and lymphoplasmacytic infiltration, small blood vessel infiltration and fibrosis, florid lymphoid follicle hyperplasia, number of IgG4(+) cells per HPF of >50, and IgG4/IgG(+) plasma cell ratio of >40%. Additionally, Case 1 showed the feature of eosinophilic abscess.

Although previous studies have shown that KD and IgG4RD are associated with the Th2-dominant immune response, their pathogeneses remain unclear. IL-4 and IL-13 secreted by Th2 cells play a role in IgE and IgG4 class switching. IL-5 can trigger IgE class switching as well as proliferation and activation of eosinophils. IL-10 secreted by Th2 and Treg cells can facilitate IgG4 class switching and inhibit IgE class switching [15,16].

KD and IgG4RD share the feature of enlarged glands or lymph nodes [13]. However, there are still some differences, and most of these features are overlapping presentations. These demographic, clinical, histological, and biochemical similarities and differences are summarized in Table 1. Compared with IgG4RD, KD shows a more prominent male dominance and wider distribution of age at onset, with the median age in the third to fourth decade, and the lumps in KD are rarely outside the head and neck region [4,5,8,13,17]. According to the 2019 ACR/EUALR criteria, IgG4RD typically affects the pancreas, salivary glands, bile ducts, orbits, kidney, lung, aorta, retroperitoneum, pachymeninges, or thyroid gland [18]. It shows a smaller overall middle-aged-male predominance, with the median age in the fifth to the sixth decade and female predominance in IgG4-related sialadenitis and dacryoadenitis [11,13,17,19,20,21].

KD and IgG4RD have abundant similarities in pathological manifestations, including follicular hyperplasia, lymphoplasmacytic infiltration, and postcapillary venule proliferation. However, they exhibit several differences. Tissue eosinophil infiltration is ubiquitous in KD but only found in approximately 50% of patients with IgG4RD [13,20,22]. Eosinophilic abscesses were observed in 74% of patients with KD [13,17], whereas only one patient with IgG4RD reported an eosinophilic abscess [13]. Moreover, IgG4(+) plasma cell infiltration can occur in KD. Hattori et al. first reported two patients with KD who showed >50 IgG4(+) plasma cells per HPF and IgG4/IgG ratios of >40% (11). Other researchers revealed that 20%–40% of patients with KD showed the presence of IgG4(+) plasma cells, which is a characteristic feature of IgG4RD. However, the occurrence of this feature in KD is less common than that in IgG4RD (*p* < 0.001) [8,13]. Storiform fibrosis and obliterative phlebitis are common pathological manifestations of IgG4RD; however, they are rarely observed in the parotid gland, lacrimal gland, lymph nodes, and lungs [23]. Further, the occurrence of both manifestations is extremely rare in KD. Only one case of KD with obliterative phlebitis has been reported to date [8]. Although every feature is not always present in KD or IgG4RD, the conditions can be distinguished through careful examination of clinical and pathological manifestations. The analysis of a specimen obtained from Case 1 only revealed the involvement of lymphoid tissue, which is not a typical organ involved in IgG4RD, as mentioned in the entry criteria of the 2019ACR/EULAR classification, although the pathological manifestations were compatible with the type II histological pattern of IgG4RD lymphadenopathy [18,24]. Moreover, the eosinophilic abscess noted in Case 1, which is a feature of KD, was not observed in Case 2. In summary, although Cases 1 and 2 shared several overlapping pathological manifestations, some key differences allowed the differentiation of KD and IgG4RD.

In cases where it is difficult to distinguish KD and IgG4RD or when any of them is still suspected, physicians tend to analyze the levels of serum eosinophil, IgG4, and IgE. Zhu et al. advised that a peripheral eosinophil level of >11.6% and serum IgG4 level of >266.5 mg/dL were ideal indicators for the diagnosis of KD and IgG4RD (eosinophil: sensitivity = 88.9%, specificity = 98.2%; IgG4: sensitivity = 86.9%, specificity = 92.9%) [14]. However, it is arduous to differentiate our two cases using this approach. Serum IgG4 levels of >135 mg/dL is one of the diagnostic criteria of IgG4RD [10,18]; therefore, if a patient is first diagnosed with KD and does not undergo immunohistological staining or serum IgG4 examination, the diagnosis of IgG4RD may be impossible.

The systemic comorbidity of KD is nephropathy [9]; however, IgG4RD affects multiple organs and may even correlate with malignancy. IgG4-related lymphadenopathy can be an initial feature or may emerge during the progression of systemic IgG4RD [25]. It remains unclear whether patients with initially localized KD will develop systemic IgG4RD. Lu et al. [26] first reported a case diagnosed with KD before being diagnosed with IgG4-related autoimmune pancreatitis and lung squamous carcinoma. However, future longitudinal studies involving more patients with clear pathological interpretations are warranted. When the coexistence and potential evolution of KD and IgG4RD are not detected, patients will only be treated with surgical excision and no systemic agents. Yet, this could increase the risk of recurrence.

Whether KD exists within the IgGRD spectrum, which includes Mikulicz disease, Küttner tumor, and Riedel thyroiditis, remains unclear. There are several diseases similar to KD and IgG4RD, such as multicentric Castleman’s disease, eosinophilic granulomatosis with polyangiitis, and lymphocyte-variant hypereosinophilic syndrome [27,28,29]. These conditions are characterized by lymphadenopathy, eosinophilic or lymphoplasmacytic infiltration, and increased IgG4(+) plasma cells. Due to these similarities, further studies examining the overlapping pathogenesis and coexistence of these diseases are warranted.

In conclusion, we report two cases with similar clinical and pathological manifestations but with different diagnoses of KD and IgG4RD, respectively. Based on their clinical similarities and different comorbidities and prognoses, both diseases should be carefully distinguished from each other. The overlapping manifestations of KD and IgG4RD and consequent effects on treatment options and prognosis should be considered.

**Table 1 jcm-11-06887-t001:** Demographic, clinical, histological, and biochemical features in Kimura disease, IgG4-related disease, and the two cases with overlapping presentations.

	Kimura Disease (KD)	IgG4-Related Disease (IgG4RD)	Case 1	Case 2
Age				
Distribution	10–80 [3,4,7,12,17]	20–80 [10,12,17,19]	38	36
Median years	30–40 [3,4,7,12,17]	50–60 [10,12,17,18,19,20]	N/A	N/A
Male predominance	Strong (80%) [3,4,7,12,17]	Slight (60%) [10,12,16,18,19,20]	Male	Male
Head and neck predominance	Strong (80%) [4,7,12]	Slight (54%) [10,12,18,21]	Postauricular region	Postauricular region
Peripheral blood eosinophilia	Frequently (80%) [4,7,8,12,17]	Occasionally (20%) [12,16,17,19,20,25]	N/A →5.8%	24.0%→19.3%
Serum IgE elevation (>100 kU/L)	Frequently (82%) [3,4,7,12,17]	Rarely (8%) [12,17]	N/A→713	N/A
Serum IgG4 elevation (>135 mg/dL)	Occasionally (11%) [3,7,12,17]	Frequently (82%) [10,12,17,19]	N/A→193	N/A→215
Histological features				
Lymphoplasmacytic infiltration	Almost always (98%) [7,12,17]	Almost always (95%) [12,17,22]	Positive	Positive
Lymphoid follicular hyperplasia	Almost always (95%) [12,17]	Almost always (92%) [12,17]	Positive	Positive
Eosinophil infiltration	Almost always (100%) [7,12]	Sometimes (49%) [12,19,22]	Positive	Positive
Eosinophilic abscess	Frequently (74%) [12,17]	Rarely (1%) [12,17]	Positive	Negative
Proliferation of postcapillary venules	Frequently (82%) [7,12]	Sometimes (55%) [12]	Positive	Positive
Stromal fibrosis	Frequently (80%) [7,12,17]	Frequently (82%) [12]	Positive	Positive
Storiform fibrosis	Rarely (6%) [7,12,17]	Sometimes present (49%) [10,12,17,22]	Negative	Negative
Obliterative phlebitis	Rarely (1%) [7,12]	Occasionally present (22%) [10,12,19,22]	Negative	Negative
IgG4 (+) plasma cell ≥ 50/HPF	Occasionally (36%) [7,12]	Almost always (100%) [12,17,19]	Positive	Positive
IgG4/IgG (+) plasma cell ratio ≥ 40%	Occasionally (39%) [7,12]	Almost always (100%) [12,17,19]	Positive	Positive
Comorbidities				
Allergic disease	Occasionally (15%) [4,7,12,17]	Sometimes (46%) [12,16,17,18,19,25]	Allergic urticaria	Negative
Nephropathy	Occasionally (14%) [4,7,8,17]	Occasionally (12%) [10,18]	Negative	Negative
Malignancy	No report	Rare (1.2%) [18]	Negative	Negative
Recurrence	Sometimes (25%) [4,7]	Sometimes (38%) [18,20,25]	Negative	Negative

## Figures and Tables

**Figure 1 jcm-11-06887-f001:**
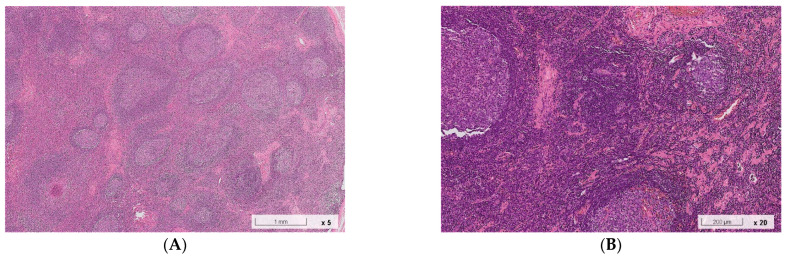
Pathological manifestations based on hematoxylin–eosin and immunohistological staining of the specimen obtained from Case 1. (**A**) Florid follicular hyperplasia, 5×. (**B**) Small vessel infiltration with fibrosis, 20×. (**C**) Stromal fibrosis, 40×. (**D**) Periventricular sclerosis, 40×. (**E**) Eosinophilic infiltration with eosinophilic abscess formation, 40×. (**F**) Immunohistological staining showed >50 IgG4(+) cells per high-power field. (**G**) Immunohistological staining showed IgG(+) cells.

**Figure 2 jcm-11-06887-f002:**
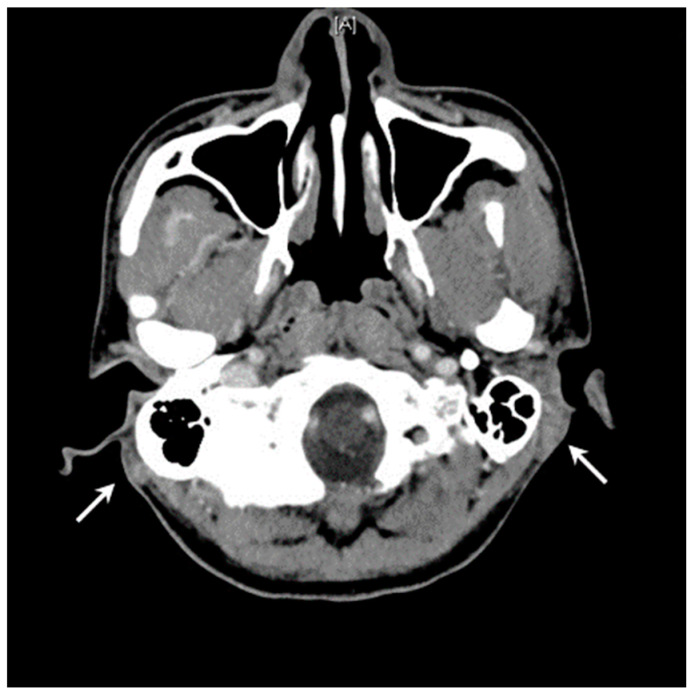
High-resolution computed tomography with head and upper neck enhancement in Case 2. The image revealed subcutaneous nodular lesions (arrows) in the bilateral postauricular region (dimensions, right: 15 mm × 7 mm; left: 17 mm × 12 mm) with relatively ill-defined margin and increased stranding of adjacent subcutaneous fat.

**Figure 3 jcm-11-06887-f003:**
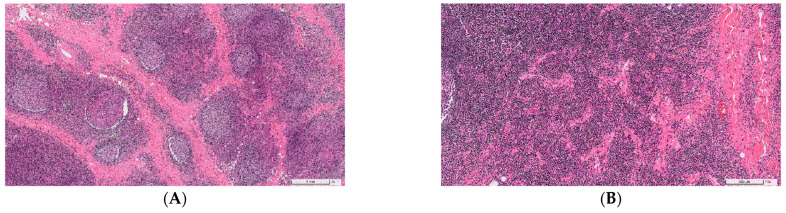
Pathological features with hematoxylin-eosin stain and immunohistology stain of the specimen from Case 2. (**A**) Foci of fibrosis, fibroadipose tissue, and salivary gland tissue with scattered lymphoid follicles containing germinal centers, 2×. (**B**) Small blood vessel infiltration. (**C**) Dense infiltration of small lymphocytes, plasma cells, and eosinophils, 60×. (**D**) Immunohistology stain showed IgG4-positive cells, more than 100 cells per high-power field (**E**) Immunohistology stain showed IgG-positive cells, percentage of IgG4-positive/IgG-positive cells was more than 40%.

## Data Availability

Not applicable.

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
