# Peer review of "Comparison of Clinical Manifestations and Pathology between Kimura Disease and IgG4-Related Disease: A Report of Two Cases and Literature Review"

_jcm, 2022, doi:10.3390/jcm11236887_

Round 1

Reviewer 1 Report

Authors carried out a nice attempt presenting the case reports and literature review on Kimura, and IgG4-related disease.  The information presented in this paper is informative to the researchers in the scientific community.  I only have few comments or suggestion.

1.      Authors are suggested to increase introduction section. Also, review the previous research findings on Kimura disease.

2.      It would be easy for readers to understand If the authors provide clinical details such as sample collection, primary, secondary end points etc.

3.        Authors are suggested to include details of instruments mentioned in this manuscript. For instance, instruments used in Immunohistology, tomography.

Author Response

Dear reviewer,

  Thank you for providing these insights to our manuscript. The suggestions and the replies are listed below:

Reviewer

Suggestion

Reply

Revision

Reviewer 1

Increase introduction section, review the previous research finding of Kimura disease.

Thank you for your suggestion. We have now revised the clinical, histological, and serological descriptions of Kimura disease, and added the pathological description of IgG4-related disease. We have also added the references which described the pathological features and possible pathophysiology of Kimura disease, and that first described Kimura disease patients with IgG4-positive plasma cell infiltration.

Kimura disease (KD) was a chronic proliferative disease that typically presents with soft subcutaneous mass in the head and neck region, focal lymphadenopathy, and lacrimal gland or sialic gland swelling. …Histology features were characterized by follicular hyperplasia, eosinophil, lymphocyte and plasma cell infiltration, and vascular proliferation. Serology test usually presented elevated serum IgE and peripheral eosinophilia (5). The pathogenesis was still unclear. However, since elevated levels of IL-4, IL-5, and IL-13 were noted in the previous study, current research consensus favored that it was related to autoimmune disease or chronic inflammation (6). …

The pathology features (of IgG4-related disease) were characterized by dense lymphoplasmacytic infiltration, storiform fibrosis, and obliterative phlebitis. However, non-obliterative phlebitis and eosinophilic infiltration could also be present. Recently, since Hattori et al. first described two KD patients with IgG4-positive plasma cell infiltration, …

Reviewer 1

Include details of instruments mentioned in this manuscript. For instance, instruments used in immunohistology, tomography.

We agree that the details of instruments should be more complete. We have now added the high resolution computed tomography images of case 2, and have clarified the examination tools, the use of contrast, and the staining method after all imaging and pathological examinations in the section of Case Presentation section.

Case 1

The patient received a C-shape incision for the postauricular mass under local anesthesia 1 month later. Pathological analysis involving hematoxylin-eosin staining revealed lymphoid tissue with eosinophilic abscess formation, … Immunopathological staining for IgG and IgG4 was also performed. Within three high-power fields (HPF) with the highest density of IgG4 and IgG-positive plasma cell, …

Case 2

Fine needle aspiration cytology was done simultaneously, …High resolution computed tomography of the temporal bone showed …Hence, the patient underwent surgical excision later. Pathology of the left lesion under the hematoxylin-eosin stain involved …

Figure 2. High resolution computed tomography with enhancement of head and upper neck from case 2. The image demonstrated subcutaneous nodular lesions in bilateral postauricular region (right: 15 mm × 7 mm; L: 17 mm × 12 mm in dimension) with relatively ill-defined margin and increased stranding of adjacent subcutaneous fat.

Reviewer 1

Provide clinical details such as sample collection, primary and secondary endpoint, etc.

This is a valid suggestion. However, both cases presented at our outpatient clinic incidentally. There was no primary or secondary endpoint. Nevertheless, we still supplemented and reconfirmed the clinical information and outcomes of the patients.

Case 2

The clinical, laboratory, and pathological features of the patient all matched the 2019 ACR/EULAR IgG4RD criteria (12). So far after the surgery, the patient has not shown any sign of recurrence

Reviewer 2 Report

Some characteristics of Kimura disease and IgG4-RD may have some similarity, however these two disorders can be distinguished by careful physical and pathological examination. Title of this manuscript must be changed. Author must describe the difference rather than similarity.

Author Response

Dear Reviewers and Editors:

Thank for your reply about “accept for Major revision”.

We give our revision and opinions as below:

Dear reviewer,

  Thank you for providing these insights to our manuscript. The suggestions and the replies are listed below:

Reviewer

Suggestion

Reply

Revision

Reviewer 2

Title of this manuscript must be changed.

Thank you for your comment. We have now changed the title to a more neutral and apposite statement: Comparison of clinical manifestations and pathology between Kimura disease and IgG4-related disease: A report of two cases and a literature review. We think this title is much better, and we hope you agree with the changes.

Comparison of clinical manifestations and pathology between Kimura disease and IgG4-related disease: A report of two cases and a literature review

Reviewer 2

Must describe the difference rather than similarity.

Thank you for your suggestion. We have now incorporated your suggestion by focusing on the pathological differences between Kimura disease and IgG4-related disease in the fourth paragraph of the Discussion section.

Tissue eosinophil infiltration is almost always present in KD, but only in about 50% of IgG4RD patients (12,19,21). Eosinophilic abscesses have been frequently seen in 74% of KD patients (12,17), while it is so rare in IgG4RD patients that only one case has been reported by Wang et al. (12). Moreover, IgG4-positive plasma cell infiltration could also occur in KD, and Hattori et al. first reported two KD patients with over 50 IgG4-positive plasma cells per HPF and an IgG4/IgG ratio >40% (11). Other researchers found that 20%–40% of KD patients met the criteria of IgG4-positive plasma cells in IgG4RD. However, it is still much less common than IgG4RD (p < 0.001) (7,12). Storiform fibrosis and obliterative phlebitis are typical pathological features of IgG4RD, but are relatively rare in the parotid glands, lacrimal glands, lymph nodes, and lungs (22). However, both of them are very rare in KD. Only one KD case with obliterative phlebitis has been reported by Kottler et al. (7). Although not every feature always presents in KD or IgG4RD, they can be distinguished from each other through careful clinical and pathological examination. Despite their similarities, we could classify them into KD and IgG4RD. The pathology sample of case 1, which only involved lymphoid tissue, was not one of the typical organs of IgG4RD mentioned in the entry criteria of the 2019ACR/EULAR classification, even though the pathological features were compatible with the type II histological pattern of IgG4RD lymphadenopathy (14,23). Moreover, the eosinophilic abscess noted in case 1 was one of the KD features, but it did not present in case 2. In summary, although case 1 and case 2 shared many overlapping pathological manifestations, there were still some differences that allowed classification into KD and IgG4RD.

Round 2

Reviewer 2 Report

Author revised the title of manuscript according to the reviewer's suggestion.

Author Response

Thanks for your positive reply.